# Oxytocin Signaling as a Target to Block Social Defeat-Induced Increases in Drug Abuse Reward

**DOI:** 10.3390/ijms22052372

**Published:** 2021-02-27

**Authors:** Carmen Ferrer-Pérez, Marina D. Reguilón, José Miñarro, Marta Rodríguez-Arias

**Affiliations:** 1Department of Psychology and Sociology, University of Zaragoza, C/Ciudad Escolar s/n, 44003 Teruel, Spain; c.ferrer@unizar.es; 2Unit of Research Psychobiology of Drug Dependence, Department of Psychobiology, Facultad de Psicología, Universitat de Valencia, Avda. Blasco Ibáñez, 21, 46010 Valencia, Spain; marina.reguilon@uv.es (M.D.R.); jose.minarro@uv.es (J.M.)

**Keywords:** oxytocin, drug addiction, social stress, corticotropin-releasing factor, reward system, animal models, human research, neuroinflammation

## Abstract

There is huge scientific interest in the neuropeptide oxytocin (OXT) due to its putative capacity to modulate a wide spectrum of physiological and cognitive processes including motivation, learning, emotion, and the stress response. The present review seeks to increase the understanding of the role of OXT in an individual’s vulnerability or resilience with regard to developing a substance use disorder. It places specific attention on the role of social stress as a risk factor of addiction, and explores the hypothesis that OXT constitutes a homeostatic response to stress that buffers against its negative impact. For this purpose, the review summarizes preclinical and clinical literature regarding the effects of OXT in different stages of the addiction cycle. The current literature affirms that a well-functioning oxytocinergic system has protective effects such as the modulation of the initial response to drugs of abuse, the attenuation of the development of dependence, the blunting of drug reinstatement and a general anti-stress effect. However, this system is dysregulated if there is continuous drug use or chronic exposure to stress. In this context, OXT is emerging as a promising pharmacotherapy to restore its natural beneficial effects in the organism and to help rebalance the functions of the addicted brain.

## 1. Oxytocin as a Promising Therapeutic Treatment for Addiction Disorders

### 1.1. Endogenous Oxytocin (OTX) System

There is huge scientific interest in the neuropeptide oxytocin (OXT) due to its putative capacity to modulate a wide spectrum of physiological and behavioral effects. It is mainly synthetized in the supraoptic nucleus (SON) and the paraventricular nucleus (PVN) of the hypothalamus, and the majority is released into the peripheral bloodstream through neurohypophysis [1]. In the periphery, OXT acts as a hormone that modulates parturition, lactation, and sexual stimulation, among other functions [2]. Within the central nervous system (CNS), oxytocinergic neurons project from the hypothalamus to a variety of brain regions such as the nucleus accumbens (NAc), prefrontal cortex (PFC), anterior olfactory nucleus, lateral septum (LS), bed nucleus of the stria terminalis (BNST), amygdala, and hippocampus [1].

The oxytocin receptor (OXTR) is a member of the rhodopsin-type 1 G protein-coupled receptor family, and the expression of these receptors is believed to fluctuate in a sex-dependent and species-specific manner [3]. These receptors are expressed in several tissues outside the CNS such as the heart, kidney, thymus, adipocyte tissue, gastrointestinal tract, mammary glands, and uterus [2]. Brain OXTRs are widely distributed and densely expressed in areas that are key to the regulation of social behavior, emotion, and motivation such as the mesolimbic circuit (including the PFC), NAc, and ventral tegmental area (VTA), among others [2,3].

This vast distribution of OXTR illustrates how OXT modulates a wide array of physiological and cognitive processes [2,4]. Considering its potential to modulate motivation, learning, emotion, and the stress response, it is a crucial component to be taken into consideration when addressing an individual’s vulnerability or resilience to developing a substance use disorder [5]. Moreover, the oxytocinergic system is reported to be altered after acute and chronic consumption of drugs of abuse, thus highlighting a possible use of exogenous OXT as a therapeutic tool in recovery from substance use disorders [6].

### 1.2. Drug Exposure Alters Oxytocin Neurotransmission

Clinical and preclinical studies suggest that oxytocinergic function in the brain is altered after acute or chronic exposure to drugs [6]. Preclinical studies show that acute exposure to psychostimulants such as cocaine can alter OXT levels in several brain areas. In a classic study, Johns, Caldwell, and Pedersen [7] found that two injections per day of 15 mg/kg cocaine over two days reduced hippocampal OXT in female rats, while no differences were observed in other structures such as the VTA or amygdala. Moreover, an up-regulation of OXTR in structures such as the piriform cortex, amygdala, NAc, and LS is usually observed after chronic psychostimulant administration in male rodents [8,9]. Interestingly, some studies have found that the dysfunction of the OXT system after psychostimulant administration can negatively affect several social and affiliative behaviors that are mediated by OXT such as pair bonding [10].

With respect to opiates, acute administration of morphine has been reported to reduce hypothalamic OXT release in lactating females [11], whereas other studies have found increased OXT immunoreactivity in the amygdala, hippocampus, and basal forebrain of male mice [12]. It has been hypothesized that these differential effects of acute opiate administration on OXT activity are dependent on the brain structure in question or the sex of the animal [13]. When considering the effects of chronic opioid administration over the OXT system, Zanos and collaborators [13,14] described different examples of hypofunction such as reduced OXT synthesis, decreased OXT plasmatic levels, decreased OXT immunoreactivity in the hippocampus, and decreased OXT mRNA levels within the SON, arcuate, and median eminence nucleus of the hypothalamus. As a consequence, a compensatory increase in OXTR binding has been observed in different brain areas such as the olfactory nuclei and the amygdala, which should be taken into consideration when deciding a possible OXT dose in subjects chronically exposed to opiates [13].

A similar pattern has been described regarding the effects of ethanol and Δ9-tetrahydrocannabinol. After chronic ethanol administration, abstinent male rats exhibit decreased levels of OXT in the brain and increased levels of OXTR in frontal and striatal areas [15], while chronic exposure to Δ9-tetrahydrocannabinol downregulates OXT-NP mRNA expression in the VTA and NAc [16].

Clinical evidence of the impact of drug consumption on the OXT system is mainly derived from two types of studies: the analysis of plasmatic OXT levels in dependent and non-dependent subjects, and post-mortem studies. Broadly, clinical studies are in concordance with the aforementioned pre-clinical results, and confirm a similar dynamic change in the OXT system. For instance, acute alcohol consumption has been shown to decrease plasmatic OXT levels in women [17], while chronic exposure commonly produces a dysregulation of plasmatic OXT levels in men [18], in addition to neuroadaptations such as increases in OXTR mRNA and binding levels in frontal and striatal brain areas in post-mortem samples from alcohol-dependent male subjects [15].

To summarize, clinical and preclinical evidence suggests that repeated exposure to drugs leads to a drop in OXT levels that seems to be related to a decrease in its synthesis [19]. While the exact mechanism that drives the decrease of OXT synthesis is not fully understood [20], the consequences of this OXT hypofunction are better ascertained and have been related to compensatory changes that upregulate OXTR in different brain areas [9,20]. These neuroadaptive changes in the endogenous OXT system mainly comprise brain regions involved in addiction and stress-related behaviors [6]. Discrepancies in the literature regarding the direction of OXT changes may be due to the brain region being analyzed [5] and the stage of the addictive cycle (acute administration, long-term dependence, or abstinence).

### 1.3. OXT Modulates the Addictive Cycle

The relationship between drugs of abuse and OXT is reciprocal. A vast body of scientific literature shows that, similar to the way drugs of abuse dysregulate the endogenous OXT system, OXT can modulate the individual’s response to drugs. The therapeutic potential of OXT has been studied in all stages of the addiction cycle [21]. It has been shown to be protective in the initial stages of addiction as it diminishes behavioral and physiological responses to drugs. Preclinical evidence suggests that OXT prevents the progression from initial experimentation with a drug to dependence and escalation of drug-taking. Finally, OXT is a promising target in the management of drug abstinence and the rebalancing of brain functions after chronic exposure to drugs. In this section, we develop the hypothesis that this therapeutic potential is due to the way OXT modulates core neurobiological systems and processes that underlie the development of substance use disorders [21] (Figure 1).

#### 1.3.1. OXT in Early Stages of Addiction

During the early stages of addiction, drug taking is motivated mainly by the acute reinforcing effects of the drug [22]. The positive sensation or pleasure that is experienced is mediated by an increase in dopaminergic activity in the mesocorticolimbic system, which is also implicated in the development of neuroadaptations that underlie context-associated memories and the attachment to incentive salience to drug-related stimuli [22]. There is clear evidence that the neuropeptide OXT interacts with the reward system, and that the rewarding effects of pair bonding, sexual contact, and social interaction depend on the actions of OXT in the mesocorticolimbic circuit [23]. Additional evidence suggests that OXT is also able to interfere with the reinforcing effects of drugs of abuse, mainly through its modulatory effect on dopamine (DA) activity in key regions such as the NAc, VTA, and PFC [24].

OXT administration has been shown to reduce drug-induced increases in DA in different mesolimbic regions including the NAc [24,25]. For instance, Sarnyai and collaborators [26] found that OXT administration (1 µg/µL) directly into the NAc blocked cocaine-induced increases in DA release in male mice. Similarly, other studies with psychostimulants reported that intracerebroventricular (2.5 μg/µL) or peripheral (1 mg/kg) administration of OXT blocked methamphetamine-induced increases in DA activation in the NAc in male rodents [27,28], an effect observed with other drugs such as alcohol. Peters and collaborators [25] found that intracerebroventricular administration of OXT (1 µg/5 µL) blocked the increase of DA in the NAc shell after ethanol administration in naïve and chronically treated male rats, and attributed the subsequent decrease of ethanol consumption in a self-administration (SA) paradigm to this effect.

There is evidence that a positive social environment plus a well-developed endogenous OXT system is a protective factor that diminishes an individual’s vulnerability to become initiated in the addictive cycle [29]. One of the mechanisms put forward to explain this effect is that OXT decreases novelty seeking and the initial response to drug reward [29,30]. Novelty seekers display a preference for novel environments and stimuli over familiar ones [31], and this behavioral trait has been linked to an increased risk of drug abuse in both humans and animal models [29,32]. For instance, rodent studies show that rats that display an enhanced locomotor response to a novel context also display an enhanced response to psychostimulant drugs and a faster acquisition of SA behavior [33]. Tops et al. [29] posited that an increased tone of endogenous OXT alters DA, serotonin, and endogenous opioid neurotransmission, thus promoting a shift in novelty processing from ventral to dorsal striatal structures and a subsequent decrease of preference and emotional reactivity to these contexts.

This protective effect of OXT in naïve animals during their first contacts with drugs has also been reported when OXT levels are acutely increased pharmacologically. For instance, the administration of intracerebroventricular OXT (2.5 μg/µL) to male mice prior to a methamphetamine conditioned place preference (CPP) protocol inhibited the acquisition of CPP for the context associated with the drug [34]. Moreover, repeated increases in OXT levels induced by pharmacological administration have been found to induce long-lasting neuroadaptive changes. In this regard, the repeated administration of OXT (1 mg/kg) during adolescence has been reported to decrease ethanol consumption and methamphetamine SA during adulthood in female and male rodents [35,36].

In addition, OXT has been shown to enhance the value of social stimuli, which in turn can act as an alternative reinforcer that changes the focus from drug reward to social reward [37]. For instance, Venniro and collaborators [38] found that in the presence of two possible rewards, social interaction versus the drug, rats that had developed methamphetamine SA behavior preferred the social reward. This effect has also been reported regarding the creation of drug-context-associated learning. For example, positive social interaction is rewarding enough to induce a strong conditioned preference [39] that can compete with the previously established conditioned preference for cocaine [40].

#### 1.3.2. OXT in the Progression to Addiction

OXT also has the potential to prevent the progression of the addictive cycle by stemming the development of dependence and the appearance of behavioral alterations after continuous contact with the drug [6,20]. The initial motivation to consume a drug is first driven by positive reward experiences; however, after repeated drug consumption, there is a shift of motivation to avoid the negative emotional state that emerges during abstinence [22].

Dependence is a state in which drug users display neuroadaptive alterations within the reward system and in other brain structures such as those implicated in the stress response [22]. OXT has been found to attenuate the development of tolerance and dependence, and this effect has also been related to its potential to modulate DA neurotransmission in the mesocorticolimbic circuit [41]. In this regard, endogenous OXT attenuates the development of tolerance to the analgesic effects of opiates, a phenomenon that has been demonstrated by intracerebroventricular administration of selective OXT antagonists to tolerant male mice, which induces a further decrease in the analgesic effect of morphine [42,43]. OXT (5 or 0.5 µg) also reduces tolerance to the sedative and hypothermic effects of ethanol [44] and modulates psychostimulant-induced stereotypy and locomotor sensitization [45]. Based on its potential to prevent the development of tolerance, it has been posited that OXT exerts a general attenuation of the neuroadaptations that underlie drug addiction [41,46].

Finally, it has been demonstrated that OXT is useful in the management of long- and short-term abstinence [21]. During abstinence, individuals generally experience negative somatic symptomatology, hypohedonia (reduced response to natural reward), stress, and anxiety, all of which drive drug seeking and consumption to relieve this negative emotional state [22]. An exhaustive description of the neurobiological mechanism that sustains the so-called “dark side of addiction” can be found in Koob and Volkow [22]. In short, this negative affect stage is the result of hypodopaminergic activity in the reward system, while there is hyperactivity in the brain stress system and hypothalamic–pituitary–adrenal (HPA) axis. Clinical and preclinical studies show that acute OXT administration can decrease both somatic and motivational symptoms of withdrawal from different drugs of abuse [21]. Generally, preclinical studies show that OXT can decrease withdrawal symptoms such as facial fasciculations, convulsion, hypothermia, and anxiety-like and depression-like behavior during morphine, cocaine, nicotine, and alcohol abstinence (an exhaustive review can be found in Bowen and Neumann [21]). For instance, when Szabó and co-workers [47] injected different doses of OXT (0.02, 0.2, and 2 IU) to alcohol-dependent male mice, they found that withdrawal convulsions decreased in a dose-dependent manner. This potential of OXT to decrease alcohol withdrawal has also been tested in a small sample of humans in a double-blind placebo-controlled trial, with positive results. Pedersen and collaborators [48] administered intranasal OXT (24 IU) twice daily for three days to alcohol-dependent patients admitted for medical detoxification. Their results showed that OXT-treated patients self-reported less severe withdrawal symptoms including craving, and required less lorazepam during their detoxification treatment. Similarly promising results have been obtained in other clinical trials with different drugs of abuse. For example, Stauffer and collaborators [49] found that intranasal OXT administration (40 IU) decreased cocaine and heroin craving in patients in methadone maintenance treatment. However, although preclinical studies provide strong evidence that OXT decreases withdrawal symptoms, clinical studies of OTX are still at an early stage, and mixed results have been reported [41].

#### 1.3.3. OXT in the Prevention of Reinstatement and Relapse into Drug Use

Three classic triggers provoke relapse into drug seeking: exposure to drug cues, drug priming, and stressful experiences. Preclinical studies highlight the potential of OXT to impede the activation of the NAc by these factors, therefore decreasing its potential to promote a reinstatement of drug seeking behavior [21]. In this regard, Cox and collaborators [50,51] found that peripheral administration of OXT (1 mg/kg) prior to drug-paired cues, pharmacological stress, or drug-priming decreased the reinstatement of drug seeking in female and male rats in a methamphetamine SA paradigm.

In human studies, a single acute dose of intranasal OTX (20 IU) has been shown to reduce the craving for tobacco induced by cues in abstinent smokers of both sexes [52], though it has been reported not to diminish the craving for tobacco induced by social stress with a high dose (40 IU) [53,54]. OXT produces mixed effects with regard to modulating craving of cannabis, which may be determined by the level of dependence; for instance, it was shown to decrease craving induced by social stress in dependent individuals [55], while no effects were observed in recreational users [56].

The anti-stress effect of OXT may be one of the mechanisms underlying its therapeutic potential. Stress is considered to be a risk factor in all stages of the addictive process, first by increasing vulnerability to drug experimentation, then by enhancing the risk of developing dependence, and finally by provoking relapse during drug cessation [57]. Additionally, physical and social stress has been demonstrated to prime the immune system into a pro-inflammatory state, which in turn has been shown to modulate vulnerability to develop several health problems including drug addiction [58]. OXT has anti-stress and anti-inflammatory effects, and also enhances the stress-buffering effects of other interventions such as social support [59,60,61]. This effect can be explained broadly due to its potential to decrease the reactivity of the HPA axis by reducing the release of adrenocorticotropic hormone (ACTH) and corticotropin-releasing factor (CRF) [1]. The importance of OXT as an anti-stress intervention is a crucial development that will be discussed in the following sections of this review.

## 2. The Anti-Stress Potential of Oxytocin

### 2.1. What Is Social Stress?

Selye [62] defined stress as a non-specific biological response of the body to a demand made on it from the outside (the environment). It is a normal adaptive response in which the body’s reserves are mobilized [63,64]. In today’s society, the main stressors that people suffer are of a social nature (social organization, social support, socioeconomic aspects, scholar life, marital status, work role, gender, discrimination, etc.). Social stress can be defined as the breakdown of the organism’s homeostasis due to neurophysiological changes derived from social and environmental events or circumstances related to the individual [65,66,67]. Inevitably, socially stressful events are part of the daily life of human beings. Sometimes these are specific events that generate a high stress response but involve a rapid recovery of homeostasis; on other occasions, the event lasts longer than the individual wishes [68,69]. When the breakdown of homeostasis is prolonged over time due to stress, there is an imbalance in the correct functioning of various brain and hormonal systems that can eventually lead to a series of diseases. Indeed, clinical and preclinical studies have demonstrated that social stress is a fundamental factor in the development of various mental disorders such as depression, anxiety disorders, neurodegenerative disorders, and addiction [70,71,72,73,74,75,76].

Studying and understanding the brain mechanisms involved in the impact of social stress on the organism is a complex issue. Animal models allow us to represent the experiences and social conditions that induce psychosocial stress in humans. In recent decades, they have facilitated an enormous expansion of our knowledge concerning social stress and the mechanisms it triggers in the stress system and other systems (such as the immune system) as well as its relationship with a large number of mental disorders. Social defeat (SD) involves an agonistic encounter between conspecifics and mimics a subordinate vs. outsider scenario in human relations [77,78]. It is considered the most representative animal model for studying social stress due to its high translational value and ecological and ethological validity [78,79,80]. The SD model consists of various agonistic encounters during which an experimental animal (intruder) is introduced into the home cage of an animal that has experienced prolonged isolation (resident), so that the resident threatens and attacks the intruder in an expression of territorial dominance [81,82]. One of the disadvantages of SD is that it is designed only for males, as female rodents are not territorial and do not engage in aggressive behavior (except under specific circumstances) and are, therefore, not sufficiently aggressive to be subject to defeat-induced stress [83,84]. To study social stress in females, alternative models are employed such as social instability or vicarious social defeat [85,86].

### 2.2. Brain Mechanisms Activated by Social Stress

The adaptive response of our body to a stressor is expressed by the activation of the HPA axis. The processing of the information of the stressor takes place in the PVN of the hypothalamus. Nerve signals promote the synthesis of CRF and other peptides such as vasopressin and OXT, which are released from the PVN, from where they are conducted to the anterior pituitary [87,88]. Once the CRF reaches the anterior pituitary, it stimulates the synthesis and release of another hormone, ACTH. This hormone is then secreted by the anterior pituitary and travels through the circulatory system until it reaches the adrenal glands, where glucocorticoids (GC) (cortisol or corticosterone) and catecholamines (adrenaline and noradrenaline) are secreted [89]. Cortisol acts in close liaison with the autonomic nervous system in such a way that if cortisol levels increase due to the effects of stress, heart rate and blood pressure also rise [90]. Moreover, GC regulate the inflammatory response through both direct transcriptional action on target genes and indirect inhibition of the transcriptional activity of transcriptional factors [91,92]. The direct transcriptional action of GC influences the induction of the synthesis of anti-inflammatory proteins such as lipocortin 1, IL-10, SLP1, or the antagonist of IL-1 receptors through the binding of GC receptors to GC response elements of DNA found in the promoter region of the target genes [93,94,95]. Indirectly, the GC receptor interacts with transcription factors such as AP-1 or NF-kB, preventing them from binding to DNA and exerting the corresponding gene regulation [93,94]. Moreover, GC receptors bind to coactivating molecules such as cAMP response element-binding protein (CREB)-binding protein (CBP) or p300, inhibiting the intrinsic activity of histone acetyltransferase (HAT) and recruiting histone deacetylases (HDAC), which promotes folding of DNA around histones. As a consequence, the transcription of pro-inflammatory proteins such as TNFα, IL-1β, IL-12, IFN-γ, IL-6, MIP-1α, MCP-1, or COX-2 is inhibited [96,97].

The HPA axis is capable of self-regulating; when blood levels of GC are high, a negative feedback is produced in the GC receptors of the hypothalamus and PVN, thereby inhibiting their secretion. This occurs due to the negative response of GC in the CRF promoter, which inhibits the binding of the transcriptional machinery to the CRF promoter, reducing the transcription of the CRF gene [98].

When this negative feedback fails, the continued activation of CRF causes a series of neuroadaptations that affect the neurotransmitter and immune systems [79]. After repeated exposure to SD, the extra-hypothalamic release of CRF activates the mesocorticolimbic DA system [99]. This activation of the dopaminergic system produces a decrease in social interaction in animals as well as depressive-like symptoms and anhedonia, amongst other effects [100,101]. The serotonin and norepinephrine systems constitute other signaling systems involved in a series of mental disorders (anxiety, depression, etc.) and are also affected by social stress [102,103,104,105]. Chronic social stress produces an upregulation of the serotonin transporter in the dorsal raphe nucleus, which translates into a deficiency of synaptic serotonin, thus contributing to the appearance of depressive symptoms that can promote the addictive process [79,106,107]. In addition, activation of the HPA axis increases the release of norepinephrine, and if this increase is prolonged, it can produce negative emotions such as anxiety and fear [108].

Prolonged exposure to social stress produces a desensitization of GC receptors [74,109,110], thus reducing the effectiveness of GCs in inhibiting transcription factors (e.g., NF-kB) and the intrinsic activity of HAT. Research suggests that chronic or prolonged social stress leads to overactivation of the immune system as a result of less suppressive effects of GC on transcription factors, and enhances expression of pro-inflammatory signals in peripheral monocytes, thus increasing inflammatory signaling [74,111,112,113].

Finally, we must mention another important effect induced by social stress; namely, the alteration of the signaling of the neuropeptide OXT. The release of OXT from the PVN has the ability to inhibit or dampen the response of the HPA axis [114,115]. As above-mentioned, OXT is synthesized by two types of neurons: parvocellular and magnocellular neurons. The main outputs of these projections feed the CRF neurons of the PVN. OXT exerts its role in the adaptive response to social stress by diminishing the stress response through the regulation of CRF. A more in-depth explanation of this modulation can be found in the next section.

### 2.3. Role of Oxytocin in Social Stress

OXT plays an important role in CRF transcription. Activation of the OXTR in the PVN interferes with the expression of CRF, and binds and sequesters the coactivator of the CRF transcription factor CREB (CRTC3). In this way, OXT reduces the binding of CRTC3 to the promoter of the CRF gene and delays its expression [116]. In this way, OXT is a stress modulator induced by activation of the HPA axis [117,118,119].

Studies in animal models and humans demonstrate this modulating effect, illustrating a reduction in the secretion of GC (cortisol and corticosterone) induced by the secretion of OXT after exposure to acute social stress [60,120,121,122,123]. In an interesting experiment, Engert and co-workers [124] observed peripheral OXT secretion in response to acute social stress (Trier Social Stress Test) in healthy humans. First, the authors associated enhanced peripheral OXT secretion with higher overall cortisol reactivity during the reactivity phase in response to psychosocial stress. Later, during the recovery phase, a greater secretion of OXT was paralleled by a more rapid recovery of vagal activity. Thus, it would appear that higher OXT plasmatic levels reduce stress reactivity faster after the stressor has ceased to be present [124,125,126].

In contrast, the consensus in the scientific literature is that when exposure to social stress is prolonged or chronic, there is a dysfunction of endogenous OXT characterized by sexual differences. Litvin and co-workers [127] reported that male mice undergoing chronic SD showed an increase in the OXTR of the medial amygdala (MeA) and LS, and hypothesized that it was involved in the negative social behaviors induced by SD. Likewise, an increase in the expression of the c-Fos protein has been observed in the OXT neurons of BNST, SON, PVN, and in the OXTRs of male mice exposed to SD [128]. Other studies have reported similar alterations in female rodents subjected to SD or social instability; for example, Steinman and co-workers [129,130] observed a long-lasting increase in OXT production and OXT/c-fos cells in the medioventral BNST and rostral PVN of female California mice, but not in their male counterparts. Hyperactivity of medioventral BNST and PVN OXT neurons in turn increases OXTR activation in the anteromedial BNST, inducing avoidance of unknown social contexts among female California mice [129,131]. Other studies have obtained contrasting results; for example, in an experiment using the chronic social instability paradigm, OXTRs were observed to be significantly increased in the amygdala and decreased in the PFC and hypothalamus of female rats [132]. Equally, female mandarin voles show a decrease in the number of OXT projections and a decrease in expression levels of proteins and mRNA of OXTR in the NAc shell after chronic SD [133].

Studies in humans have shown that people with major depression disorder (MDD) faced with a social exclusion task exhibit lower plasmatic levels of OXT than controls [134], which produces an increase in negative emotions in these circumstances. This can be interpreted as an endogenous deficit that leads to isolation and personal failure [135], or alternatively, as a protective mechanism against social stress by which lower levels of OXT regulate aversive social cues so that social stress stimuli are experienced as less aversive [136]. Gender differences have also been described, since women with chronic depression and MDD show lower plasma levels of OXT than males [134,135]. Decreased plasma levels of OXT have also been observed in patients with social anxiety disorder [137], which could reflect a lower ability to react pro-socially [138].

To summarize, both animal models and human studies have revealed unquestionable discrepancies concerning the effects of social stress on endogenous OXT that involve sex differences. It should be noted that testosterone can reduce the activity of the HPA axis, while it has been shown that estrogens can increase or reduce the activity of this axis according to the differential activity on the estrogen α (increased activity of the HPA axis) and β (decreased HPA axis activity) receptors [2,121,132]. We can conclude that social stress modifies the projection of OXT neurons and alters plasmatic levels of OXT in both humans and animals. These changes produce a dysfunctionality in the main role of OXT in stress, which is the modulation of the HPA response and reduction of the impact of the stress response.

### 2.4. Exogenous OXT Administration as a Therapeutic Target for Social Stress Disorders

Based on the above-mentioned results, a large number of studies have been carried out to explore the administration of OXT as an agent for treating depressive or anxiogenic mood states derived from social stress exposure. These studies have also highlighted important discrepancies, due perhaps to varying forms of administration and doses employed and gender differences. For example, Eckstein and colleagues [136] observed that healthy men who received intranasal OXT (24 IU) before social stress (The Montreal Imaging Stress Task) did not show increases in cortisol levels in saliva, but did manifest an increase in perceived social stress. In contrast, Heinrichs and co-workers [60] reported the opposite, registering protective effects in healthy men who received the same intranasal dose of OXT before the Trier Social Stress Test. In said studies, the subjects did not display alterations in the HPA or the OXT system as a result of prolonged or continuous exposure to social stress. Other studies have been conducted in patients with a history of depression, anxiety disorders, or prolonged exposition to social stress during childhood. In male patients with generalized anxiety disorder, intranasal administration of OXT (24 IU) normalized the decreased connectivity of the frontal-amygdala (left and right amygdala with rostral anterior cingulate cortex/medial PFC (mPFC) connectivity) that usually occurred in these subjects during the emotional face processing task [139,140,141,142]. Additionally, these patients showed increased empathy after receiving intranasal OXT (24 IU) in a reward motivation task [143]. The benefits of treatment with OXT have also been observed in male subjects who have experienced early adverse experiences, in whom improvements were observed in the connectivity between the amygdala and the PFC [144] and in the ability to face emotion recognition [145]. In a more recent study, the efficacy of intranasal OXT (24 IU) in improving the recognition of emotions has also been demonstrated in healthy adults of both sexes [146]. Additionally, in a series of studies carried out in male and female police officers with post-traumatic stress disorder, intranasal administration of higher doses of OXT (40 IU) improved motivation tasks and social reinforcement. Likewise, through functional magnetic resonance imaging (fMRI), an increase in neural responses was observed in the striatum, dorsal anterior cingulate cortex [147], and insula [147,148] with respect to patients treated with a placebo. Several studies by Koch and co-workers suggest that OXT acts as a cognitive regulator of emotions and anxiety; for instance, it decreased anxiety before an emotional faces recognition task and lowered the reactivity of the amygdala during the task [149]. In an extension of the study in question, an increase in connectivity between the left thalamus and amygdala was observed during a distraction task in male patients [150].

Similarly, preclinical research has revealed that injections of OXT (1 ng/200 nL) into the NAc shell reverses the alterations in social behavior, anxiety, and depressive symptoms induced by chronic SD, while injections of an antagonist blocks the effects of OXT in female mandarin voles [133,151]. Local administration of OXT (1 or 5 µg) to the PFC of male mice can reduce depressive-like symptoms in defeated mice, with a subsequent increase in DA levels taking place. Moreover, OXT increases phosphorylation of protein kinase A (PKA) and dopamine and cAMP-regulated phosphoprotein of 32 kDa (DARPP-32) in intracellular PKA/DARPP-32 signaling dependent on D1 receptor activation. Thus, the enhancement of dopaminergic transmission induced by administration of OXT reduces the depressive symptoms produced by SD in the mPFC [152].

## 3. Oxytocin Blocks Increased Drug Intake Induced by Social Defeat

SD stress induces long-lasting changes in the reward system that affect the response to drugs of abuse such as cocaine or ethanol. After SD, increases in the acquisition of and motivation to take cocaine has been extensively reported using the SA procedure [153,154,155,156]. Equally, defeated mice show higher ethanol intake in the oral SA paradigm [61,76,157,158]. Studies using the CPP paradigm have also shown an enhancement in the conditioned rewarding effects of cocaine and ethanol in stressed animals [73,158,159,160,161]. Both the aforementioned paradigms permit a broad assessment of the rewarding effects of drugs of abuse as both measure the role of motivation and environmental cues [162,163].

Repeated social stress and prolonged substance use induce long-lasting changes in the reward system that influence both processes mutually, since a state of chronic stress will produce neuroadaptations that promote the development of the addictive spiral, and vice versa [164,165]. Together, OXT and DA play a crucial role in the reward system; DA in the reward system areas, VTA and NAc are negatively affected by exposure to continuous stress and by continued consumption of substances of abuse, both of which decrease DA [161,166,167,168]. As previously mentioned, OXT is released as an adaptive factor in the face of acute stress, but its synthesis can be impaired when stress is prolonged [115,169]. Additionally, after prolonged drug use, the OXT system induces neuroadaptive changes in areas involved in the reward system (VTA, NAc) that produce an upregulation of OXTR [6,9,19,20,170]. Moreover, OXTRs are expressed by non-dopaminergic neurons in the VTA such as GABA and glutamate neurons, which can modulate the activity of these DA VTA neurons locally (in opposite directions) or project to other brain regions including the NAc where they can alter the positive reinforcement or aversion produced by substances of abuse [171,172,173].

Taking into account the scientific evidence presented so far, OXT is a promising therapeutic target that reverses the effects of social stress (Table 1). With the aim of testing this therapeutic potential, our research team has conducted various studies to explore the role of OXT in relation with social stress and addictive behavior. First, we evaluated social housing conditions as a possible protective factor against the negative effects induced by SD such as the increased rewarding effects of cocaine or anxiety-like behaviors. Stressed mice were housed in five different conditions: standard housing (four males in each cage); two males in a cage from adolescence or starting in adulthood; and one adult male housed with a female for a short or long period. Male mice that had lived in an environment with quality social attachments (paired with a female) or with another male from adolescence onwards displayed a resilient profile after social stress experiences, with decreased response to cocaine reward and less anxiolytic responses [174]. This protective effect correlated with increased plasmatic OXT levels in the mice housed with a female. Although levels also increased in defeated males paired with another male from adolescence onwards, they did not reach the levels observed in those paired with a female. However, blockade of endogenous OXT using the OXTR antagonist atosiban (1 mg/kg) completely annulled the protective effect of housing conditions in the case of the co-housed males, thus highlighting a critical role for OXT [175].

If favorable social housing can avoid the increase in the rewarding effects of cocaine induced by SD through an increment of OXT, we predicted that the same results could be obtained if we administered exogenous OXT to animals housed under standard conditions. To test this hypothesis, we administered, peripherally, one dose of OXT (1 mg/kg) before each agonistic encounter. In relation to cocaine, our results showed that the administration of OXT before each SD blocked the long-term increase in the conditioned rewarding effects of cocaine and cocaine SA induced by SD. We also observed that OXT was capable of undermining the reinstatement of cocaine-seeking behavior in the latter paradigm. In addition, OXT administration prior to each SD decreased anxiety-like behaviors [59].

Finally, we set out to test if this beneficial effect of OXT could also be applied to the increased ethanol intake observed in socially stressed mice. Our results confirmed that peripheral administration of OXT (1 mg/kg) before each SD reversed the long-term negative effects of SD on ethanol consumption [61]. OXT-treated stressed animals consumed similar amounts of ethanol to the control group and showed less motivation to obtain the drug compared to untreated defeated animals in the oral ethanol SA paradigm.

Other groups have evaluated the protective role of OXT against relapses into the consumption of different drugs of abuse as a result of social stress. OXT (1 mg/kg) attenuated alcohol-seeking behavior in a dose-related manner in male and female mice in response to an acute challenge with a predatory odor [176]. Similar results have been observed in the tempering of drug seeking and reinstatement due to different types of stress. For example, intracerebroventricular OXT (0.1, 0.5, 2.5 μg/µL) attenuated the reinstatement of restraint stress-induced methamphetamine CPP [34], and systemic administration of 6.4 mg/kg of carbetocin (OXT analog) reduced the effects of forced swimming stress on CPP reinstatement induced by morphine [14] in male mice. Furthermore, systemic administration of OXT (1 mg/kg) decreased methamphetamine-seeking behavior after exposure to a predator’s odor [177] and administration of yohimbine in rats [50]. In humans, intranasal OXT (40 IU) may be beneficial in reducing the risk of relapse among cocaine-dependent individuals (both sexes) with a history of mistreatment during childhood [178]. Another study in male military veterans with post-traumatic stress and alcohol use disorder revealed that intranasal administration of OXT (40 IU) attenuated the reactivity of cortisol to stress tasks, though it did not mitigate craving [179]. In adult tobacco smokers who have suffered adverse experiences in childhood, intranasal administration of OXT (40 IU) prior to the Trier Social Stress Task has been shown to attenuate cortisol response in both sexes. However, this reduction in cortisol was larger in men with greater childhood adversity, pointing to a greater benefit from the anxiolytic properties of OXT in this sex [180].

Several mechanisms could explain the above-mentioned beneficial effect of OXT. One factor to take into account is that OXT can modulate the neuroinflammation induced by SD (Figure 2). Numerous reports confirm that social stress induces a potent and long-lasting neuroinflammatory response [181,182,183,184]. Neuroinflammation is composed of a series of cellular (affecting mainly microglia) and molecular (increased release of cytokines and chemokines) alterations in the form of an immune response within the CNS. Neuroinflammation is not only provoked by pathological conditions, but is also trigged by psychological stress (see review [182]). SD-induced neuroinflammation provokes increases in macrophages and activated microglia in the brain [185], thus increasing pro-inflammatory cytokines and chemokines (as IL-1β, IL-6 or CX3CL1) [61,157,159,174,186]. In addition, exposure to SD augments the permeability of the blood-brain barrier, allowing immune cells to cross into the CNS [183]. OXT is reported to inhibit pro-inflammatory mediators such as TNF-α, IL-1β, and nitric oxide synthase [187,188,189]. We have recently confirmed these effects when we observed that peripheral OXT (1 mg/kg) administration mitigated the neuroinflammatory response induced in male mice by repeated SD. These OXT-treated animals displayed neuroinflammatory levels of chemokines (CX3CL1 and CXCL12) that were lower than those observed in non-treated defeated mice [61]. Similar results have been obtained with other types of social stress. For example, intracerebroventricular administration of OXT (1 μg/µL) in adult male mice subjected to maternal separation mitigated the increase in the expression of genes relevant to the neuroinflammatory response (IL-1β, Myd88, TNF-α, TLR4, and Nlrp3) [190]. Moreover, intranasal administration of OXT (1 μg/µL) to male rats reversed the increase in IL-1β and IFN-γ levels in the mPFC and hippocampus induced by a rodent model of posttraumatic stress disorder [191].

## 4. Final Remarks

The preclinical and clinical studies reviewed in this paper clearly supports the hypothesis that endogenous and exogenous OXT exerts a protective effect in all the stages of the addiction cycle by modulating the initial response to drugs and by attenuating the development of dependence. In this regard, the anti-stress potential of the neuropeptide is especially promising as it buffers against the negative impact of stress over drug-related behaviors.

However, the studies referred to in the present review have inherent limitations that are worth taking into consideration such as the existence of sexual differences in OXT dynamics, both in humans and animal models. Moreover, there are methodological differences between studies that can be summarized in the following aspects: (1) the paradigms used to provoke social stress vary among the different studies; (2) different methods of administering OXT-peripheral, intranasal, or intracerebral-have been used and will have undoubtedly exerted a different effect in each study; and (3) there is no unanimity with respect to the doses used in animal models. Bearing this in mind, we can affirm that the study of the role of OXT in social stress and the addictive process is at an early stage, but despite these limitations, it should continue based on the promising results obtained thus far.

This review sheds light on the importance of the role of the oxytocinergic system as a fundamental component in the understanding of substance use disorder. However, continuous drug use or chronic exposure to stress dysregulates this system. Exogenous OXT administration is a potentially interesting pharmacotherapy that allows the natural beneficial effects of OXT to be reaped and the balance in the functions of the stressed/addicted brain to be restored [192].

## Figures and Tables

**Figure 1 ijms-22-02372-f001:**
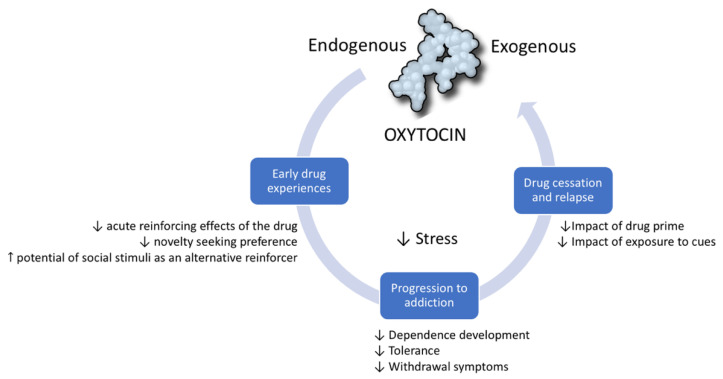
Oxytocin modulates the addiction cycle. Clinical and preclinical studies show that oxytocin has protective effects during all stages of the addiction cycle: (1) In early drug experiences, it diminishes the reinforcing and general effects of drugs. (2) It can prevent progression from initial drug experimentation to dependence and drug escalation. (3) In drug cessation, it can prevent relapse induced by cues, drug primes, and stress.

**Figure 2 ijms-22-02372-f002:**
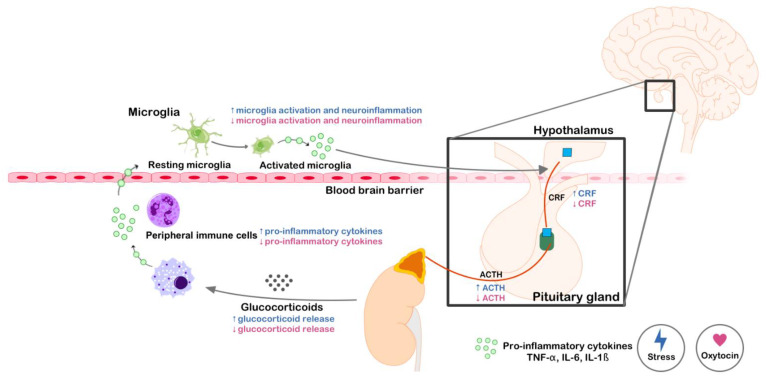
Oxytocin modulates the inflammatory response induced by stress. The effects of stress are depicted in blue, and those of oxytocin in pink. Stress activates the hypothalamic–pituitary–adrenal (HPA) axis and promotes the release of glucocorticoids that turn peripheral immune cells into a primed state. The activation of stress-primed immune cells can induce an exaggerated inflammatory response that eventually reaches the brain and promotes the activation of resident immune cells (neuroinflammation). Activated microglia within the brain release inflammatory cytokines that activate the HPA. Oxytocin counteracts the effects of stress in all the stages of this loop.

**Table 1 ijms-22-02372-t001:** Effect of the administration of oxytocin as a therapeutic target in the addictive process induced by social stress in animal models and clinical studies.

Species	Stress Type	Paradigm/Drug Disorder	Treatment/Dosage	Finding	Reference
Male mice	Social Defeat	Cocaine CPP	Oxytocin 1 mg/kg; i.p.	Blocked the long-term increase in the conditioned rewarding effects, favored extinction and decreased reinstatment	59
		Cocaine SA			
Male mice	Social Defeat	Ethanol SA	Oxytocin 1 mg/kg; i.p.	Blocked the long-term increase ethanol consumption and seeking behavior	61
Male and female mice	Predator odor	Ethanol SA	Oxytocin 1 mg/kg; i.p.	Attenuated alcohol seeking behavior	176
Male mice	Restraint	Methamphetamine CPP	Oxytocin 0.1, 0.5, 2.5 μg/μl; i.c.v.	Attenuated reinstatment	34
Male mice	Forced swimming stress	Morphine CPP	Carbetocin * 6.4 mg/kg; i.p.	Attenuatted reinstatment	14
Male rats	Predator odor	Methamphetamine CPP	Oxytocin 1 mg/kg; i.p.	Decreased CPP acquisition	177
Male and female rats	Yohimbine	Methamphetamine CPP	Oxytocin 1 mg/kg; i.p.	Decreased CPP acquisition	50
Male and female humans	Mistreatment during childhood	Cocaine-dependent	Oxytocin 40 IU; intransal	Reduced risk of relapse	178
Male military veterans humans	Post-traumatic stress disorder	Alcohol use disorder	Oxytocin 40 IU; intransal	Attenuated reactivity of cortisol to stress task, but did not reduce craving	179
Male and female humans	Adverse experiences in childhood	Tobacco smokers	Oxytocin 40 IU; intransal	Attenuated cortisol response in trier social stress task. Greater anxiolytic effect in men	180

* Carbetocin is a synthetic analog of oxytocin.

## Data Availability

Not applicable.

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
