# Peer review of "Oxytocin Signaling as a Target to Block Social Defeat-Induced Increases in Drug Abuse Reward"

_ijms, 2021, doi:10.3390/ijms22052372_

Round 1

Reviewer 1 Report

The manuscript presented faces a topic of high relevance for the scientific community. The manuscript is well written and the cited literature is appropriate. However  few changes might improve the quality of the manuscript.

Please address the following comment:

1) Paragraph 1.2 This paragraph lacks of a possible explanation of the effects observed (such as increased oxytocin immunoreactivity, increase of brain oxytocin or oxytocin receptor etc etc). What is the behavioral consequence of such neurochemical observations? 

2) Line 71-73: it would be useful to explain the differences observed in the different brain areas in relation to the "specific mechanisms" in which those brain areas are involved

3) In all the manuscript: the authors often refer to the Nucleus Accumbens, but they never specify between Core and Shell subnuclei. Since dopamine in these nuclei plays different role in the behavior and addiction, it would be useful to specify the implication of dopamine core and shell when in the text the author refer to the Nac

For example see reference : doi: 10.1016/s0166-4328(02)00286-3 doi: 10.1016/j.neuropharm.2020.108176

4) In all the manuscript the authors not specify the dosage of oxytocin used in the preclinical studies and few times they specify the dosage used in the clinic. Adding the dosage (at least the range and the route of administration) is useful for a better interpretation of the data (for example compare the dosage used with the physiological endogenous levels of oxytocin, or oxytocin levels following a stimulus that is known to increase the oxytocinergic system (social interaction for example)).

5) Moreover it would be important to specify the sex of the animals or humans and take in consideration that the oxytocinergic system is affected by estrogens.

6) A table that summarize the results described in the text, regarding each specific stimulus (drugs, alcohol, stress) will be useful for the readers.

Reviewer 2 Report

The review of Ferrer-Pérez et al. covers a vast literature regarding the interaction of stress, addiction and their impact on oxytocinergic system. The manuscript is clear and well-structured in each section and the argument fits perfectly with the topic. My suggestion to further improve the quality and impact of the manuscript is to re-elaborate the final remarks session, which I feel a bit unbalanced towards the limitations in the field (which obviously are correct). I suggest the authors also stress out also the strengths oxytocin as a successful therapy for stress-induced addiction worsening maybe with encouraging conclusion and/or perspectives.

Minor points:

-Please, revise line 503 (.. to take into account is that that OXT can modulate..) that is repeated twice

-Please, revise that the OXT acronym appears always instead of the whole name once it has been described previously (i.e. line 330, OXT instead of oxytocin)

Author Response

We completely agree with the reviewer that the “Final remarks” section is unbalanced towards the limitations of the reviewed studies. Following the reviewer’s suggestion, we have rewritten this section highlighting the positive conclusions regarding the potential of the oxytocinergic system as a therapeutic target. Now it reads as follows (Lines 563-585):

The preclinical and clinical studies reviewed in this paper clearly supports the hypothesis that endogenous and exogenous OXT exerts a protective effect in all the stages of the addiction cycle by modulating the initial response to drugs and by attenuating the development of dependence. In this regard, the anti-stress potential of the neuropeptide is especially promising as it buffers against the negative impact of stress over drug-related behaviors. 

However, the studies referred to in the present review have inherent limitations that are worth taking into consideration, such as the existence of sexual differences in OXT dynamics, both in humans and animal models. Moreover, there are methodological differences between studies that can be summarized in the following aspects: 1) the paradigms used to provoke social stress vary among the different studies, 2) different methods of administering OXT - peripheral, intranasal or intracerebral - have been used and will have undoubtedly exerted a different effect in each study; and 3) there is no unanimity with respect to the doses used in animal models. Bearing this in mind, we can affirm that the study of the role of OXT in social stress and the addictive process is at an early stage, but despite these limitations, it should continue based on the promising results obtained so far. 

This review sheds light on the importance of the role of the oxytocinergic system as a fundamental component in the understanding of substance use disorder. However, continuous drug use or chronic exposure to stress dysregulates this system. Exogenous OXT administration is a potentially interesting pharmacotherapy that allows the natural beneficial effects of OXT to be reaped and the balance in the functions of the stressed/addicted brain to be restored [192].

Regarding the typographical mistakes stressed by the reviewer on his “minor points” comments, we have addressed all the issues. In line 530, we have eliminated the repeated “that” and we have also revised that the acronym OXT always appears in the body of the manuscript instead the whole name (Changes in lines 35 and 341).